# A qualitative study of healthy eating, physical activity, and mental health among single mothers in Canada

**Lisbeth A. Pino Gavidia**[1]*, **Joy C. MacDermid**[1,2], **Laura Brunton**[1], **Samantha Doralp**[1]

1 Department of Health and Rehabilitation Sciences, Faculty of Health Sciences, Western University, London, ON, Canada, 2 Roth McFarlane Hand and Upper Limb Centre, St. Joseph's Health Care London, London, ON, Canada

☯ These authors contributed equally to this work.

* lpino2@uwo.ca

**Data Availability Statement:** All relevant data is uploaded within Supporting Information files.

**Funding:** The authors received no specific funding for this work

## Abstract

Single motherhood is associated with higher demands in home and family responsibilities that may exist in similar sized households with grandparents. These demands can compete with opportunities and resources for maintenance of healthy behaviours. This may have been exacerbated during the COVID-19 pandemic since supports outside the home may have been restricted by public health measures. A qualitative narrative study was conducted to explore these issues with 12 single mothers in Canada. Semi-structured interviews provided an in-depth understanding of the challenges and facilitators to maintaining healthy eating habits, physical activity levels, and mental health. Theory-driven content and structural analysis were applied through a descriptive thematic approach. Challenges to maintaining healthy eating behaviours included stay-at-home orders, limited budget, unhealthy food/cravings, and lack of motivation. In contrast, living with grandparents who provide help or having an understanding of healthy food were factors that facilitated engagement in healthy eating. Challenges to maintaining physical activity levels included lack of willpower, lack of time, and low energy. Whereas time on one's own, weighing scales or outdoor gear, and weather conditions promoted physical activity. Spirituality and gratitude were the main coping mechanisms used to deal with loneliness, unemployment, and depressive symptoms. Further research into the utility of gratitude as a coping mechanism could inform public health interventions that aim to increase levels of well-being among single mothers.

## Introduction

There has been an increase in the number of families headed by single parents since the late twentieth century [1]. Increased rates of divorce have resulted in single-parent families, particularly headed by women, and it is fairly common that most children live permanently with the mother [2]. In Canada, single parenthood is mostly experienced by women [3]. These changes in the family structure impact the health and well-being of women; for example, single

**Competing interests:** The authors have declared that no competing interests exist.

mothers under the age of 50 are more vulnerable to experience poor health and disability compared to married mothers [4]. In addition, single mothers often face financial strains because they are the lone income provider which limits their ability to purchase nutritious food through socially acceptable means. Families with low incomes are more vulnerable to the consumption of less healthier diets [5, 6]. One earner and lower wages often result in low-income households and poor nutrition.

In the context of Canada, a lower intake of fruits and vegetables is observed among single mothers [2, 7]. This often occurs because women prioritise the needs of their children before their own [8]. In other words, mothers sacrifice their nutrition to maintain their children's diets by modifying their own food consumption [9]. For instance, 52% of single mothers deprived themselves of good nutrition to meet the needs of their children in a low-income community of Ontario, Canada [9]. Inadequate income restricts healthy eating practices for mothers who are working at a minimum wage [5].

The literature on women's health has focused on their multiple roles either in the labour force or in the family responsibilities [10]. Being a single mother who is raising children on one's own requires a full-time commitment to family roles. Caring for children is one of the family role responsibilities that is time-consuming among single mothers [6]. Research showed that time available for grocery shopping, preparation, and good nutrition is in jeopardy due to time pressures at home and family responsibilities [8]. Single mothers experienced time scarcity, leading to high consumption of energy-dense fast foods and low consumption of fruits and vegetables [8]. Food preferences also influences eating behaviours, which accounts for sensory appeal, price, and convenience [5, 11]. Food choices are influenced by a variety of factors varying from individual to social and cultural factors.

Due to an overload in these roles, a higher proportion of stress-related outcomes occur, including higher levels of depression and anxiety in single mothers compared to married/partnered mothers [10, 12]. The presence of depression contributes to low levels of physical activity, and poor health [4]. Physical activity is considered one of the public health strategies to help people improve their health [12]. However, around 60–80% of adults are not physically active enough to achieve health benefits [13]. Public health guidelines for physical activity suggest moderate-intensity aerobic activity for $\geq$ 150 minutes every week, combined with muscle-strengthening activities on at least for two days per week [4, 14]. Overall, women experience low rates of physical activity, particularly single mothers with young children [13].

In Western societies, physical activity often happens during leisure time [15]. Mothers who are working full-time and raising children on their own may not have enough access to leisure-time activities [16]. In turn, single mothers juggle multiple family role responsibilities often prioritizing family needs and sacrificing self-pursuits to do so [17]. Despite the recognition that physical activity optimizes health and well-being, family roles and busy schedules are barriers for single mothers to engage in leisure-time physical activity [13, 18]. This may also be due to conflicts associated with family life and work.

Family responsibilities fall disproportionately among women globally [19]. During the COVID-19 pandemic, single mothers faced further imbalance between home and work, often being most responsible for caring for children in the home [20]. Single mothers experienced high levels of parenting stress and depression during the pandemic [21]. Increased parenting stress has been demonstrated to be associated with low levels of physical activity [4]. In addition to stress and depression, working mothers experienced high levels of loneliness and isolation during the COVID-19 pandemic [21]. High stressors include social isolation and lack of support that add into the demands of daily life [22, 23]. This highlights that single mothers experience constraints at many levels, which may compromise their mental health.

Due to overload in family responsibilities and work, single mothers in industrialized societies have poor mental health [10]. Depression is the major mental illness among women with the prevalence of depression being two to three times higher in single mothers compared to the general population [24, 25]. Worldwide estimates show that 9.5% of women experience depression compared to 5.8% of men [26]. In particular, depression disproportionally affects the maintenance of healthy eating and physical activity in which single mothers are more likely to experience depression than married mothers [26]. For instance, if single mothers have more stressors, they have a higher probability of experiencing persistent levels of depression over 1-year period [24]. This may demonstrate that single mothers are a more vulnerable group relative to the rest of the population.

Single mothers report resilience through coping mechanisms in day-to-day living. Resilience means to bounce back and adapt in response to adversity [27]. All people are born with the capacity to develop resilience traits such as a sense of purpose, problem solving, and social competence [27]. Resilience helps single mothers to cope with difficult situations. Coping is a cognitive and behavioral strategy to control emotional and depressive symptoms, and interacts with sense of belonging [23, 25]. Coping is also a positive predictor for women's mental health [25]. In turn, coping strategies have two types. First, emotion-focused coping is to manage emotions. Second, problem-focused coping is to deal with problems [28]. There are few studies that examine which coping strategies are used by single mothers. The identification of mechanisms that help single mothers to cope and function would benefit their health and well-being.

Most studies have targeted specific issues or subgroups of single mothers. For instance, areas in the following domains: African American low-income single mothers [22]; employment status, welfare issues, and absence of support sources [29]; economic deprivation [27]; comparison between single-mother families to two-parent families [22]; and life satisfaction and psychological distress [23]. Relatively few articles have captured how single mothers maintain their health, juggle to adapt family responsibilities, and support their children's health as they follow COVID-19 guidelines. More research is needed to identify how single mothers implement healthy behaviours and what things help or prevent those changes with a focus on Canadian women who are parenting alone. We ask the following research questions, recognizing that both systemic and pandemic-related issues might arise: 1) What are the experiences of single mothers as they try to maintain healthy eating, physical activity, and mental health? and 2) What are some of the challenges and facilitators to maintain those healthy behaviours? Qualitative interviews provided in-depth understanding on the challenges and needs of mothers' health and well-being.

## Materials and methods

This is a qualitative study in which single mothers aged 18–50 were recruited through purposeful sampling seeking variation in age and employment status. Participants had to meet all of the following specific criteria. First, participants were Canadians who self-identified as single mothers and spoke fluent English. Second, participants were over the age of 18, and had dependent children living in the home. Finally, participants were able to provide informed consent. Semistructured interviews were used to facilitate conversation with the aim to explore participants' lived experiences while investigating the research questions (see S1 Appendix).

Two main principles of an adequate sample size are suggested to reach thematic saturation [30]. First, specify an initial sample size upon which is the first round of analysis. In this view, the initial analysis sample yielded 12 interviews, using a process of constant comparison in which saturation occurred. Second, identify a final criterion if needed to be further conducted

if new ideas are not emerging. Data was transcribed verbatim and transferred to NVivo 12 software to facilitate coding and analysis of participants' responses.

## Study design

This study used a narrative inquiry methodology to advance the understanding of the experiences of single mothers. To engage in narrative inquiry, the authors become co-participants in knowledge co-creation alongside the participants [31]. Co-participants acknowledge the co-construction of the research-participants relationship and the importance of engaging participants in research process. This study was conducted with the interpretative paradigm where knowledge construction embodied individual lived experiences of what participants perceived to exist about their personal health [32].

## Study procedures

This qualitative study provided a rich description of the lived experiences of single mothers regarding their beliefs and lifestyles on healthy behaviours. It also examined the unique factors that help or prevent healthy eating, physical activity, and contribute to mental health. All study procedures were reviewed and approved on May 17, 2021, by the Western University Health Science Research Ethics Board (HSREB), Project ID 118854.

Participants were recruited through a poster using the official Facebook and Twitter social media platforms of the Hand and Upper Limb Centre (HULC). Then, participants to volunteer for the study contacted the main author by email in which the letter of information and signed consent were provided through a Western Qualtrics online form prior to the one-on-one interviews. As a result of the COVID-19 pandemic, interviews took place over a video/call using Western Corporate Zoom and a semi-structured interview guide. Participants were asked about healthy eating, physical activity, and mental health to maintain general health, and how easy and difficult was to implement those healthy behaviours. All data collection occurred with the same interviewer between May and July 2021.

Content and structural analyses were conducted on the interview transcripts. Content analysis was intended to identify emerging codes from the transcripts that characterized the data [33]. Sentences were coded and then compiled into categories to establish themes. Structural analysis was then used to interpret the meanings of the narratives as a whole towards the identification of themes [33].

## Results

Demographic information in Table 1 was collected from twelve single mothers. Nine single mothers had school age children, whereas three single mothers had preschool age children in daycares. Seven single mothers were unemployed and had one child, while five single mothers were employed and had two children. Out of the twelve single mothers, two identify their ethnicity as Korean, two as Indian, and the remainder as Canadian. The primary and most time-consuming family responsibility for single mothers was caring for children in the home during COVID-19.

The research data revealed in-depth insights of the experiences of single mothers about their family responsibilities and healthy behaviours. The themes that emerged from the data were the overarching connection between the codes in Table 2.

**Table 1. Characteristics of study participants and their family responsibilities.**

| ID | Age | Number of children | City | Occupation status | Family responsibilities |
|---|---|---|---|---|---|
| (1) | 39 | 1 school age daughter | St. Thomas | Unemployed | Care for the child in the home |
| (2) | 45 | 2 school age children; daughter and son | London, ON | Employed | Care for the children in the home |
| (3) | 29 | 1 preschool age son in daycare | London, ON | Unemployed | Care for the child in the home |
| (4) | 18 | 1 preschool age daughter in daycare | Chatham Kent | Unemployed | Share family responsibilities with grandparents, and care for the child in the home |
| (5) | 41 | 2 school age children; daughter and son | London, ON | Unemployed | Care for the children in the home |
| (6) | 51 | 1 school age son | St. Thomas | Unemployed | Care for the child in the home |
| (7) | 40 | 2 school age sons | London, ON | Employed | Care for the children in the home |
| (8) | 37 | 2 preschool age sons in daycare | London, ON | Unemployed | Care for the children in the home |
| (9) | 29 | 1 school age daughter | Caledonia | Unemployed | Share family responsibilities with grandparents, and care for the child in the home |
| (10) | 45 | 1 school age daughter | Masstown | Employed | Care for the child in the home |
| (11) | 38 | 2 school age daughters | Newcastle | Employed | Care for the children in the home |
| (12) | 33 | 1 school age son | St. Thomas | Employed | Care for the children in the home |

## Affordable foods were not necessarily healthy: It is an individual issue, but it is also a societal issue

Single mothers had knowledge about healthy eating behaviours, but stress and busyness prevented them from preparing healthy homemade meals. To be the only adult in the house was a stressful situation that led single mothers to grab something quick and simple rather than cook. Some mothers opted for meal kits that were delivered to the house which came with healthy ingredients. Single mothers reused leftovers during the week as a strategy to make healthy foods more affordable. Cooking is time-consuming, and single mothers had limited time as they fulfill family and work expectations. One participant mentioned, *"We are so busy nowadays, and a lot of people aren't eating home cooked meals"* (ID 9, 29 years old, 1 child). This trend led to consuming easily accessible fast foods: hamburgers, fries, and pizza. Single

**Table 2. Barriers and facilitators of healthy behaviours among single mothers.**

| Theme—Healthy Eating | | |
|---|---|---|
| Affordable foods were not necessarily healthy: it is an individual issue, but it is also a societal issue | **Barriers**<br>• Stay-at-home orders<br>• Limited budget<br>• Unhealthy food/cravings<br>• Lack of motivation | **Facilitators**<br>• Living with parents who help<br>• Understanding of healthy food |
| **Theme—Physical Activity** | | |
| As single mothers were burdened with stress; their physical activity was mostly walking and online video exercises amid the pandemic | **Barriers**<br>• Lack of willpower<br>• Lack of time<br>• Low energy | **Facilitators**<br>• Time on one's own<br>• Weighing scales or outdoor gear<br>• Weather conditions |
| **Theme—Mental Health** | | |
| Spirituality helped single mothers the most to move on after negative events | **Barriers**<br>• Loneliness<br>• Unemployment<br>• Depressive symptoms | **Facilitators**<br>• Spirituality<br>• Gratitude |

mothers were aware of not eating too much of these fast foods because of its negative health impacts. Although cooking was not an enjoyable activity, single mothers expressed the need to prioritize the nutrition of their children with vegetables. One participant noted:

> *I think I worry more about with my daughter that I do with myself, which is not necessarily good. Today, for example, I cut up vegetables on plate to go with her and I just eat the pizza. (ID 1, 39 years old, 1 child)*

To maintain a sustainable approach to healthy eating, single mothers tried to establish flexible food principles that could support positive health behaviours. One participant remarked:

> *When you are trying to be too rigid, you end up setting yourself up for failure. I learned that the hard way. If you say I am not going to have any carbs, you set yourself to the point that you just give up. So, I really like this 80/20 principle where the 80% of the time, we are eating really healthy, which I wanted to do anyway. (ID 11, 38 years old, 2 children)*

Flexibility was an important factor and can be seen in the 80/20 principle discussed by this participant. For example, 80% of the time families eat healthy foods and 20% of the time they allow themselves to have less healthy foods as a treat. This approach was more sustainable for life rather than mothers feeling deprived when trying to be 100% healthy. Single mothers were mindful of the consequences of an unhealthy lifestyle associated with food, that might contribute to changes later in their older age.

**Barriers of healthy eating.** The stay-at-home orders associated with the COVID-19 pandemic affected eating schedules, in particular lunch time because of children's online home-schooling. One participant stated, "*It is very challenging because their lunch break is 10:50 to 11:50. Who has lunch at 10:50 in the morning?" (ID 7, 40 years old, 2 children)*. There was a need for more meals and more "stocking up" behaviours as mothers and children were at home more. This increased the financial burden and often there was not enough budget to cover this increased demand. One participant remarked, "*What am I going to buy? what recipes do I want to make? I spend all my money . . . and we don't really have enough budget for everything." (ID 6, 51 years old, 1 child)*. Most of the mothers' budget was spent on food, especially vegetables and meat. The predominant barrier was that single mothers did not find the motivation to cook. One reason was being the only adult in the household, as well as not having extra time. One participant commented:

> *I used to eat a sandwich for lunch every day, and it was really fast to make it and it was really fast to eat. Now, I have a salad to make and it takes longer to make and longer to eat. It just takes extra time. I am still having to cook even when I am really tired. (ID 10, 45 years old, 1 child)*

At the end of the day, single mothers were exhausted due to the extensive family responsibilities performed within the home. Mothers recognized the need to learn time management skills to handle time in an effective way.

**Facilitators of healthy eating.** Due to social isolation that single mothers experienced during COVID-19, a few moved back to their parents' house, which in turn facilitated healthy eating. One participant said, "*My mom is good at cooking dinner. She does not deprive everybody in my family for a decent meal" (ID 9, 29 years old, 1 child)*. Single mothers had knowledge that healthy food maintains their own personal health. Small steps led single mothers to great changes. One participant mentioned, "*My parents are really good at maintaining that . . . We*

*have change from white bread to whole wheat. That white bread change still doesn't sit it right with me somedays. [But] I am getting there." (ID 4, 18 years old, 1 child).* Single mothers expressed willingness to undertake healthy behaviours. For example, to invest more time on new menus as their children are growing up. One participant remarked, *"I read a lot what boys need. I asked the doctor recently because he is growing." (ID 7, 40 years old, 2 children).* Single mothers supported healthy eating for their children, and were willing to be well and healthy for their children.

### As single mothers were burdened with stress; Their physical activity was mostly walking and online video exercises amid the pandemic

To manage stress, single mothers recognized the need to be physically active. Outdoor exercises were the mostly valued physical activities by the majority of participants in our study. One participant accounted, *"Everything impacts you, your mental health, getting out for a walk, getting fresh air and exercise. That is so beneficial." (ID 3, 29 years old, 1 child).* In particular, walking became a priority among single mothers during lockdown measures since it was perceived as convenient, inexpensive and safe. During COVID-19, single mothers faced loneliness at home, which negatively impacted their mental health. However, the pandemic encouraged them to spend time walking in the surroundings of nature. One participant explained:

> *Before COVID-19 there were so many things competing for attention. We do live in a very much consumer society. It is that pursue of happiness through things and through work, which a lot of people are still working. It is very hard to have fun without spending money, and when everything is closed. You have to find different ways to entertain yourself. (ID 9, 29 years old, 1 child)*

As there were less options and things to do during the pandemic, going back to basics like walking encouraged single mothers to be creative about being physically active.

Walking was the gentle approach implemented by single mothers. One participant recalled, *"A lot of times, I've been hard on myself. . . Some week ago, I did HIIT exercises, so High Intensity Interval Training. It is more of an intense, kind of a crazy thing. My body was not happy after. I just trying to be gentle to myself" (ID 3, 29 years old, 1 child).* Intense exercises ended up in exhaustion. As a result, it was important for single mothers to find more moderate types of physical activity. Walking was carried out from 2–4 times every week for at least one hour.

**Barriers of physical activity.** Single mothers were mindful that physical activity must be incorporated into their regular routine for their health and well-being. However, the practice of physical activity on a regular basis was linked with willpower. One participant illustrated, *"I don't want to do because I am not the person who likes exercise. Every morning, I feel lazy, I do not want to do that." (ID 6, 51 years old, 1 child).* Contradictory attitudes toward physical activity were exposed. While other participants expressed the importance of developing self-disciple regardless the circumstances. One participant pointed out, *"It is usually 8 or 9 o'clock at night. It is always at night. I know that this has to happen for my wellness, so I make that happen." (ID 11, 38 years old, 2 children).* Single mothers were responsible for all family tasks. By the time, they had time to exercise, they were extremely tired. One participant added, *"I think again just being tired. I feel like my resources are limited." (ID 3, 29 years old, 1 child).* As such, mothers had to manage and prioritize their limited resources between home responsibilities and their own personal health.

**Facilitators of physical activity.** Single mothers felt comfortable to do physical activity outside the home and without their children. One participant reflected, *"I exercise by myself*

*without my children. Just going for walks because they will not do it for one hour or two hours. So, we have different habits." (ID 5, 41 years old, 2 children).* As gyms were closed during the COVID-19 pandemic, walking became the main outlet to escape the pressures of family responsibilities. Weight scales were a tool for single mothers to check their weight regularly, and a facilitator for walking. As such, walking turned into a personal motivation towards achieving positive health behaviours. This was related to 'why' single mothers were exercising. For example, one participant said, *"When I see my body, I think that I have to do exercise" (ID 5, 41 years old, 2 children).*

Equipment was considered necessary to going outside. One participant stated, *"I got a special insulated running jacket and insulated waterproof shoes. That helped a lot. Instead of paying for the gym membership, I invested in the outdoor gear." (ID 10, 45 years old, 1 child).* The summer season was also an important factor that single mothers took advantage of. One participant highlighted, *"My biggest exercise right now is getting out with my daughter, walking down the park, see the dogs, the playground, the splash pad. In the summer, we often do lots of different zoos. It is mostly walking" (ID 1, 39 years old, 1 child).* In fact, weather conditions facilitated outdoor activities for both walking and recreational activities.

## Spirituality helped single mothers the most to move on after negative events

Single mothers described that the first shutdown was very challenging because they spent a lot of time with their children. One participant said, *"I don't think I was ever born to stay-at-home mom. I've loved this time with my daughter, and gets to be a lot some days." (ID 1, 39 years old, 1 child).* The mental health of single mothers was compromised. One participant expressed, *"Mental health is the problem. I feel very bored and stressful with things to do and to stay home every day." (ID 5, 41 years old, 2 children).* The overload of family responsibilities overwhelmed single mothers. Over the years, several of the single mothers reported to have been battling depression. Participants attributed their depression to unhealthy relationships with their ex-partners and suggested that social and emotional support from friends was important to their mental health. Once participant commented:

> *It took me a really long time to leave that relationship. I think 9 years . . . It was one of the hardest times in my life to make that decision and not looking back to that. I have a friend who literally said, you are picking a day and you are leaving. She was there, she helped me. Honestly, I don't know if I wouldn't have done it without her really helping me. (ID 11, 38 years old, 2 children)*

Some single mothers embarked on a journey of spirituality to improve their mental health. Spirituality was an anchor in their lives. Single mothers pointed out that their relationship with God gave them hope to face adversity. One participant expressed, *"I try to do for my mental health is just take some time to sit and pray and read the Bible." (ID 10, 45 years old, 1 child).* Another participant accounted, *"God helps me. I still have my core values and beliefs that I learned. There are still there. I know He is there. I have the knowledge that Jesus is my savior and saves me every day." (ID 8, 37 years old, 2 children).* To have the compass of living a life in the spirit helped single mothers to move forward in a journey of healing and gratitude.

**Barriers of mental health.** As the sole adult in the home, single mothers experienced loneliness. One participant reflected, *"My biggest thing is that as far as being a mom is that some days I have to figure all out. Nobody is going to do it if I don't do it." (ID 1, 39 years old, 1 child).* There were many uncertainties regarding the pandemic. One participant said, *"There is*

*no way out of it, just to wait, and we are still waiting.*" *(ID 10, 45 years old, 1 child)*. Worries about unemployment and children's school crowded their minds. Additionally, limited contact with their loved ones triggered isolation, which contributed to exacerbation of their depressive symptoms. One participant remarked, *"I will call my grandma and just cry. I just call my dad, and I just cry. What is wrong with me. What do I feel this? I feel such a burden or heaviness."* *(ID 3, 29 years old, 1 child)*. It was difficult for single mothers to realize that depression was happening again.

**Facilitators of mental health.** Spirituality was raised as a support for multiple participants. A source of encouragement was Scripture, which helped single mothers to find peace that God was with them in the midst of adversity. Spirituality was an important area that mothers explored during the pandemic, and gratitude was enacted from spiritual disciples. One participant stated, *"I am a religious person. I hold on to the Bible verses, which means a lot to me, and this has proved maybe from my childhood."* *(ID 2, 45 years old, 2 children)*. Gratitude made single mothers more resilient to enter the process of healing from depressive symptoms. Gratitude gave them hope in giving thanks for things they have taken for granted in their lives. One participant reflected:

> *Some days, it just be, I have breath in my lungs. . . I just walk out my door and go for a walk in a safe country where I have free fluent water. Some of these things. When you start thinking like that it changes everything about your day and your whole mentality. (ID 11, 38 years old, 2 children)*

## Discussion

This study found that single mothers experienced substantial challenges in maintaining healthy behaviours due to competing demands, economic limitations, fatigue, and lack of social supports. These challenges were aggravated during the pandemic as employment, family and social supports, school resources and contributions to childcare were compromised. A recurrent guiding principle pre- and post-pandemic was for mothers to place their children's needs first. Living with grandparents, gratitude, spirituality, exercise and time in nature mitigated the stress, despair, and loneliness felt by mothers and helped them engage in positive health behaviours.

Single mothers encountered limited time and budget hindering them from making homemade quality meals, leading them to access less expensive industrial foods high in sugar, salt, and fat, but lower in nutritional value. This was also connected to food culture within the Canadian context such that mothers recognized that industrial and fast foods are pleasure oriented, as well as require little effort to prepare. Research shows that psychological factors drive behaviours and food choices such as the sensory appeal of smell, taste, and appearance, and is considered the most important parental motive for food choices [2].

Single motherhood was related to lower consumption of vegetables. Being the only adult in the household influenced lower intakes of fruit and vegetables due to fulfilling family role responsibilities. A self-reported lack of time hindered mothers' ability to make personal healthy choices, even if they had adequate knowledge about healthy behaviours. However, this did not always extend to their children's eating behaviours. Our findings were consistent with another study that women from multi-adult households experienced fewer challenging situations than women who live alone with their children [34]. Maternal food self-depravation happened among lone mothers as they prioritized family responsibilities, including feeding their

children with healthy food [9]. Living with grandparents helped single mothers to eat healthy compared to those living in single-adult households. This may demonstrate that family structure is related to burden of family responsibilities, such that in multi-adult households the responsibilities are shared among other household members.

The main stressors discussed in this research were unaided childbearing and lower levels of support among single mothers. Lack of time and low energy was the most frequently reported barrier. This was consistent with another qualitative study that reported high levels of stress among parents with young children [6]. Competing family responsibilities between home and work roles represent time constraints for physical activity among single mothers [12, 18]. Before COVID-19, single mothers may have been physically active by spending energy on caregiving and housework. Research shows that activities that involve childcare and housework may maintain caloric balance, but are not considered sufficient for cardiorespiratory fitness [16]. During COVID-19, single mothers took the initiative to do online-based exercises and outdoor walking. This finding confirms what another qualitative study found, that walking was the primary leisure-time physical activity of single mothers [4]. The pandemic motivated some single mothers to spent time in nature four times per week for about an hour, and occasionally follow online video exercises.

Although external supports were lost during the pandemic, single mothers felt comfortable to go walking on their own, while their school-age children stayed at home. Single mothers acknowledged that physical activity has a positive impact on their health and well-being. Intrinsic motivation was an important value of mothers' long-term adherence, which encouraged physically active roles by setting a schedule until a habit is developed. It is necessary to understand the psychological motivation to participate in physical activity. There is substantial research that suggest that self-motivation is a requirement to maintain physical activity [16, 18]. In this study, access to weight scales, outdoor gear, and favourable weather conditions were also identified as important facilitators for physical activity. Finally, flexibility to adapt to busy schedules will be a recurrent need among single mothers after the pandemic to pursue physical activity.

Single mothers experienced loneliness during the COVID-19 pandemic. Feelings of isolation aggravated symptoms of depression as their primary focus was to take care of their children in the home. Another contributing factor to their depression was broken relationships with their ex-partners. Research revealed that the dissolution of a relationship or marriage is often accompanied by depression [1]. Our study highlighted that difficulty in past relationships had negative consequences on mothers' mental health. While supportive relationships benefited mental health through reassurance, assistance, and positive interactions [22]. Yet single mothers from our study tended to have fewer or no emotional and tangible supports after dissolution of relationships.

Coping mechanisms may moderate the influence between stressors and depressive symptoms [22, 23]. In this study, spirituality and gratitude were coping mechanisms that single mothers implemented to move forward after negative events of life. Single mothers explained that the uncertainties of the pandemic were the anchor to return to faith. In turn, single mothers embarked on a spiritual journey to also heal emotional wounds from past relationships. As result, gratitude arose as an expression of not taking things for granted such as the gift of life and the presence of loved ones. "Gratitude is a life orientation toward noticing and appreciating the positive effect in life" [22, p. 6]. In this study, gratitude nurtured the strength of mothers' hearts and was related with less depressive symptoms, positive associations in the levels of personality traits, and relationships with others. Spiritualty and gratitude were important constructs of mother's mental well-being that should be explored further in future research.

## Implications

Healthy behaviours cannot improve unless we provide sufficient support to single mothers in ways that are in line with their needs, including childcare and emotional support. Research showed that perceived social support is an important factor within resilience [22]. For healthy eating habits, food kits might be useful for time management by reducing the need to plan meals and shop for ingredients and have a benefit of potentially increasing cooking skills as they come with step-by-step instructions. Single mothers discussed needing self-discipline and a gentle approach to participate in physical activity as evidenced by undertaking daily walks or following online video exercises. It is recommended to plan physical activity on moderate intensity levels and regular leisure time routines. Single mothers will be more likely to plan physical activity if they feel more confident in their ability to engage in physical activity and would lessen the stressors of home and family responsibilities. During the COVID-19 pandemic, it will be important to continue these habits as life changes post-pandemic. Experiences of resilience highlighted the importance of incorporating spirituality and gratitude as components for future individual and community-based health interventions for single mothers. Spirituality became a resiliency tool used by single mothers as a coping mechanism to face unprecedented events of life. Future research on health interventions for single mothers should consider extrapolating the finding to other developed countries, as well as incorporating components of spirituality, gratitude and positive emotions with the goal of increasing health and well-being.

## Limitations

As this study was conducted during the COVID-19 pandemic (May-June 2021), single mothers had sufficient time to participate in the one-on-one interviews. While this study successfully addressed diversity through number/age of children, employment status, ethnicity, and mothers' ages, it also may have suffered from cultural bias and small samples size effects. Most of the findings are applicable to a wide pool of mothers such as being stuck at home with children, outside walks, online video exercising. Future research may repeat or extend this similar study, but include a wider pool of mothers (see S1 Appendix). However, the COVID-19 pandemic is over and the opportunity to collect similar data is also over. Thus, this study may be applicable within a different context when mothers are stuck at home due to unemployment, poverty, etc.

## Conclusion

Single mothers faced unique challenges associated with home and family work role responsibilities. Participants struggled to maintain their own personal health due to the multiple family responsibilities they perform on a daily basis, particularly caring for children in the home during the COVID-19 pandemic. Time pressures compromised mothers' healthy eating, physical activity, and mental health. Both time constraints and the ease of accessing processed foods challenged mothers' ability to make healthy choices. Mothers' physical activity was outdoor walking and online video exercises that positively impacted unaided childcare, financial strains, and unemployment. Although the pandemic exacerbated symptoms of depression and feelings of isolation, walking contributed with the prevention of sedentary behaviours and stress. Spirituality and gratitude were identified as key coping mechanisms to moderate psychological distress. Further studies should focus on how gratitude supports mothers' mental health and well-being

## Supporting information

**S1 Appendix. Interview topic guide.**
(DOCX)

## Author Contributions

**Conceptualization:** Lisbeth A. Pino Gavidia, Joy C. MacDermid, Laura Brunton, Samantha Doralp.

**Formal analysis:** Lisbeth A. Pino Gavidia, Joy C. MacDermid, Laura Brunton, Samantha Doralp.

**Methodology:** Lisbeth A. Pino Gavidia, Joy C. MacDermid, Laura Brunton, Samantha Doralp.

**Project administration:** Lisbeth A. Pino Gavidia.

**Supervision:** Joy C. MacDermid, Laura Brunton, Samantha Doralp.

**Validation:** Joy C. MacDermid, Laura Brunton, Samantha Doralp.

**Writing – original draft:** Lisbeth A. Pino Gavidia, Joy C. MacDermid, Laura Brunton, Samantha Doralp.

**Writing – review & editing:** Lisbeth A. Pino Gavidia, Joy C. MacDermid, Laura Brunton, Samantha Doralp.

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
