## [Decision Letter · Decision Letter 0]

3 Jan 2023

PONE-D-22-11799A qualitative study of healthy eating, physical activity, and mental health among Canadian single mothersPLOS ONE

Dear Dr. Pino,

Thank you for submitting your manuscript to PLOS ONE. After careful consideration, we feel that it has merit but does not fully meet PLOS ONE’s publication criteria as it currently stands. Therefore, we invite you to submit a revised version of the manuscript that addresses the points raised during the review process.

We look forward to receiving your revised manuscript.

Kind regards,

Ali B. Mahmoud, Ph.D.

Academic Editor

PLOS ONE

Journal Requirements:

Reviewers' comments:

Reviewer's Responses to Questions

**Comments to the Author**

1. Is the manuscript technically sound, and do the data support the conclusions?

Reviewer #1: Yes

Reviewer #2: No

2. Has the statistical analysis been performed appropriately and rigorously? 

Reviewer #1: N/A

Reviewer #2: N/A

3. Have the authors made all data underlying the findings in their manuscript fully available?

Reviewer #1: Yes

Reviewer #2: Yes

4. Is the manuscript presented in an intelligible fashion and written in standard English?

Reviewer #1: Yes

Reviewer #2: Yes

5. Review Comments to the Author

Reviewer #1: Many thanks for the opportunity to review this very interesting paper. It has real strengths but needs some more work to improve it. I have made more detailed comments below.

Introduction

Your justification for the research could be strengthened to really emphasise why this particular research study was needed.

Methods

Could you give more detail on how participants were recruited eg. how did you contact them? Where did you get their contact details? Etc.

Did you aim to recruit participants from different ethnic groups?

In table 1 the ‘city’ column seems to indicate some mothers were from the UK but the title state ‘Canadian single mothers’- please clarify.

A separate section on ‘Analysis’ is needed and more detail on this is needed. You state the analysis is ‘theory-driven’ but do not say how or which theory/ theories are being used. You state the authors ‘become co-participants’ but more detail is needed on what exactly this refers to.

Did you use NVivo or any other software to manage the analysis?

Please provide a reference for ethical approval.

Results

The results are an interesting read and the quotes really illustrate the hardships the participants are facing.

Overall, the results are too long and could be more succinct and less repetitive in places.

The subtitles are too long and should be more succinct. It would be helpful if the subtitles were the same as the 3 themes in table 2.

Be careful not to overgeneralise your results eg. rather than reporting your findings as ‘Canadian single mothers…’ or ‘Single mothers’, say ‘Participants….’

Discussion

The discussion repeats some of what is written in the results but needs to focus more on how the findings relate to the other literature.

Your discussion should have a section on the strengths and limitations of the study (Eg. Sample, methods, data quality eg did online interviews impact the data quality?)

Recommendations

I think the recommendation of providing food kits is good. Is there anything specific that could be recommended in relation to physical activity?

I wish you the best of luck with your revisions.

Reviewer #2: The topic is interesting and timely. The presentation style is cohesive and easy to follow. However, I have several issues with the current version of the study that potentially could be reconciled if authors consider reworking the paper.

My issue with the current version of the study is that I am not convinced that it contributes anything novel to the literature. While the focus of the study is clearly on single mothers and Canadians, neither of those two characteristics seem to play a critical role in formulating the obtained results. I think most of the findings apply to non-single mothers stuck in quarantine situations at home with their kids. For example, things like “going back to the basics (line 334)” in reference to starting outside walks are applicable to a wide demographics of people who are stuck at home under pandemic closures. Same is true with exercising using online videos etc. Small sample size of 12 is not helping here either as it is not convincing.

Being a Canadian vs American or for that matter from any other developed country doesn’t seem to matter in the content of the survey performed. If fast food is easily available and fruits and vegetables could be purchased, should one want to engage in healthy eating, (which is typical for any developed country), the findings could be extrapolated to other developed countries (at least in recommendations/future research section).

I suggest authors should think about generalizing the outcomes of the study to generate a novel enough content to warrant publication. My suggestion is to cut down on quotes from the respondent’s surveys as those are too lengthy for the value they contribute. I do not find much value in those quotes. Instead, I would like to see literature on behavioral outcomes including for example eating habits resulting from stress (as it could be applicable here), including during prolong conflicts such as war.

I suggest adding research limitation section where the small sample size should be addressed.

I believe if authors re-focus this paper on unique contributions to the literature of this study instead of simply describing the study, this paper could became publishable.

Additional grammar errors to correct

Line 111 delete “ of resilience”

Line 144 change transfer to transferred

Line 247 change access to accessible

Line 263 change thy to they

6. PLOS authors have the option to publish the peer review history of their article (what does this mean?). If published, this will include your full peer review and any attached files.

Reviewer #1: No

Reviewer #2: No

---

## [Author Response · Author response to Decision Letter 0]

6 Mar 2023

Attached, please find the document 'Response to reviewers' for the changes made.

---

## [Decision Letter · Decision Letter 1]

2 Jun 2023

PONE-D-22-11799R1A qualitative study of healthy eating, physical activity, and mental health among single mothers in CanadaPLOS ONE

Dear Dr. Pino,

Thank you for submitting your manuscript to PLOS ONE. After careful consideration, we feel that it has merit but does not fully meet PLOS ONE’s publication criteria as it currently stands. Therefore, we invite you to submit a revised version of the manuscript that addresses the points raised during the review process.

We look forward to receiving your revised manuscript.

Kind regards,

Ali B. Mahmoud, Ph.D.

Academic Editor

PLOS ONE

Reviewers' comments:

Reviewer's Responses to Questions

**Comments to the Author**

1. If the authors have adequately addressed your comments raised in a previous round of review and you feel that this manuscript is now acceptable for publication, you may indicate that here to bypass the “Comments to the Author” section, enter your conflict of interest statement in the “Confidential to Editor” section, and submit your "Accept" recommendation.

Reviewer #2: (No Response)

2. Is the manuscript technically sound, and do the data support the conclusions?

Reviewer #2: Yes

3. Has the statistical analysis been performed appropriately and rigorously? 

Reviewer #2: N/A

4. Have the authors made all data underlying the findings in their manuscript fully available?

Reviewer #2: Yes

5. Is the manuscript presented in an intelligible fashion and written in standard English?

Reviewer #2: Yes

6. Review Comments to the Author

Reviewer #2: Comments/observations

This is an improved version of the paper and I appreciate your efforts in trying to incorporate the reviewer’s feedback. I still find issues with the way paper is presented and I summarized most of them below.

1) I still find that the quantity of quotes offered in sections following table 2 are excessive for the value they bring to the paper. I do not find them to be insightful. I would like to see fewer quotes. I would rather see themes and percentages of respondents who mentioned a particular topic and/or responded to a particular theme, than a quote from one person. Alternative way to handle quotes could be creating an appendix and putting all quotes into the Appendix and just referring the reader to the Appendix if the reader wants to see a particular quote.

2) Abstract: Lines 24 and 33: you refer to “dual parent families” in line 24 but you also refer to “parents” in line 33 as grandparents of a child. Please sort out this confusion as it is not clear in the abstract before the reader knows anything about your study whether you are referring to single parent or multi-parent household. I would suggest calling mother’s parents everywhere in the paper as grandparents. It will avoid any confusion.

3) Line 62: I do not see that it was illustrated before that single mothers are employed in low-paying jobs. This is taken out of context. Please provide references or create context so that it is clear that you are looking at the low-income community, for instance.

4) Line 187: replace “as the remainder” with “and the remainder.”

5) Table 1: why are some responsibilities stated as providing emotional support to a child and some are not? What does it mean when a mother does not have this responsibility? Does it have a significant impact on the outcomes?

Also, some responsibilities are listed as shared with parents. Whose parents? Mother’s? This is a perfect example of the continued confusion of using the same “parent” term for mother and grandparents. Please consider changing to grandparents.

Please clarify all these nuances, as it is not clear if these differences make a significant impact on the results. If these are not relevant, then please use the same language to describe duties of each mother. If these are significant differences, then please offer some discussion.

6) Lines 214-215: Sentence “However, …” Please rewrite this sentence to indicate that single mothers were not eating junk foods because they were aware of negative health impacts of such foods.

7) Lines 500 – 503: You need to expand your process of thinking here because it is not clear what you mean. Why is the fact that findings can be generalized to non-single mothers is a limitation? Shouldn’t it be a strength? Please elaborate on every strength and every limitation.

8) This entire added section “Strength and Limitations” needs to be re-written as it reads as perfunctory text with not meaning. It is not enough to acknowledge a limitation/shortcoming by stating it’s existence You also need to offer a strategy of how to deal with it in another study, or in a future research. For example, why is your sample so small? What can you do differently in the design of another study to generate a larger sample? Perhaps you can say that next study should include all mothers, not only singles. You need to offer practical strategies for tackling listed limitations.

7. PLOS authors have the option to publish the peer review history of their article (what does this mean?). If published, this will include your full peer review and any attached files.

Reviewer #2: No

---

## [Author Response · Author response to Decision Letter 1]

7 Jul 2023

Thank you very much for your comments. We incorporated all the feedback into the manuscript as well as include the changes in the document 'response to reviewers'

---

## [Decision Letter · Decision Letter 2]

4 Oct 2023

PONE-D-22-11799R2A qualitative study of healthy eating, physical activity, and mental health among single mothers in CanadaPLOS ONE

Dear Dr. Pino,

Thank you for submitting your manuscript to PLOS ONE. After careful consideration, we feel that it has merit but does not fully meet PLOS ONE’s publication criteria as it currently stands. Therefore, we invite you to submit a revised version of the manuscript that addresses the points raised during the review process.

We look forward to receiving your revised manuscript.

Kind regards,

Ali B. Mahmoud, Ph.D.

Academic Editor

PLOS ONE

Journal Requirements:

Reviewers' comments:

Reviewer's Responses to Questions

**Comments to the Author**

1. If the authors have adequately addressed your comments raised in a previous round of review and you feel that this manuscript is now acceptable for publication, you may indicate that here to bypass the “Comments to the Author” section, enter your conflict of interest statement in the “Confidential to Editor” section, and submit your "Accept" recommendation.

Reviewer #2: (No Response)

2. Is the manuscript technically sound, and do the data support the conclusions?

Reviewer #2: Yes

3. Has the statistical analysis been performed appropriately and rigorously? 

Reviewer #2: N/A

4. Have the authors made all data underlying the findings in their manuscript fully available?

Reviewer #2: Yes

5. Is the manuscript presented in an intelligible fashion and written in standard English?

Reviewer #2: Yes

6. Review Comments to the Author

Reviewer #2: This is a much-improved paper. It reads well. My only suggestion is to rework the Strengths and Limitations section as it is still disjoint and weak compared to the rest of the paper. I offered my additional suggestion below.

Page 7; Section Materials and Methods

Complete the following sentence: “First, specify an initial sample size upon which is the first round of analysis.”

Table 1:

For entry 10 is stated as preparing meals while others do not have this as listed responsibility. Please either remove this or add to other mothers for consistency in describing their responsibilities.

Page 18; Section Barriers of mental health

Delete “were exacerbated” from the following sentence: “Additionally, limited contact with their loved ones triggered isolation, which contributed to exacerbation of their depressive symptoms were exacerbated.”

Page 23: Section Strengths and Limitations

This is the weakest section and needs additional work as it reads disjointly.

I suggest perhaps changing this section to Limitation only.

Within the text, you can then say that while your study successfully addressed diversity, etc., it also may have suffered from cultural bias, small sample effects, etc….

The following sentence should be modified:

“Most of the findings applied to mothers in general such as being stuck at home with children, outside walks, online video exercising.”

What are you trying to say here? Many of the finding from this study are applicable to a wide pool of mothers? Or that the findings from this study are generalizable to a wider selection of mothers? Please clarify. If that is the case, then you can say that future research may want to extend/repeat similar study but include a wider pool of mothers. However, Covid pandemic is over and the opportunity to collect similar data is over. Thus, perhaps you can suggest a different context when mothers are stuck at home (unemployed, poor, etc.).

Statements relating to small sample size do not make sense in the context of repeating the same study for all mothers:

“Finally, the sample size was small. To tackle all of these limitations, the next study should include all mothers, not only single mothers.”

Pandemic is over, thus recreating the same conditions will not be feasible. You need to come up with similar circumstances to suggest recreating a similar study.

7. PLOS authors have the option to publish the peer review history of their article (what does this mean?). If published, this will include your full peer review and any attached files.

Reviewer #2: No

---

## [Author Response · Author response to Decision Letter 2]

2 Nov 2023

The responses are outlined in the document named `response to reviewers`

Thanks.

---

## [Editor Report · Decision Letter 3]

5 Nov 2023

A qualitative study of healthy eating, physical activity, and mental health among single mothers in Canada

PONE-D-22-11799R3

Dear Dr. Pino,

We’re pleased to inform you that your manuscript has been judged scientifically suitable for publication and will be formally accepted for publication once it meets all outstanding technical requirements.

Kind regards,

Ali B. Mahmoud, Ph.D.

Academic Editor

PLOS ONE
---

## [Editor Report · Acceptance letter]

15 Nov 2023

PONE-D-22-11799R3 

A qualitative study of healthy eating, physical activity, and mental health among single mothers in Canada 

Dear Dr. Pino Gavidia:

I'm pleased to inform you that your manuscript has been deemed suitable for publication in PLOS ONE. Congratulations! Your manuscript is now with our production department. 

Kind regards, 

on behalf of

Dr. Ali B. Mahmoud 

Academic Editor

PLOS ONE